# Determination of Self-Heating in Silicon Photomultipliers

**DOI:** 10.3390/s24092687

**Published:** 2024-04-24

**Authors:** Erika Garutti, Stephan Martens, Joern Schwandt, Carmen Villalba-Pedro

**Affiliations:** Institute for Experimental Physics, University of Hamburg, Luruper Chaussee 149, 22761 Hamburg, Germany; stephan.martens@desy.de (S.M.); joern.schwandt@desy.de (J.S.); carmen.villabla@desy.de (C.V.-P.)

**Keywords:** SiPM, radiation damage, self-heating

## Abstract

The main consequence of radiation damage on a silicon photomultiplier (SiPM) is a significant increase in the dark current. If the SiPM is not adequately cooled, the power dissipation causes it to heat up, which alters its performance parameters. To investigate this heating effect, a measurement cycle was developed and performed with a KETEK SiPM glued to an Al_2_O_3_ substrate and with HPK SiPMs glued to either an Al_2_O_3_ substrate or a flexible PCB. The assemblies were connected either directly to a temperature-controlled chuck on a probe station, or through layers of materials with defined thermal resistance. An LED operated in DC mode was used to illuminate the SiPM and to tune the power dissipated in a measurement cycle. The SiPM current was used to determine the steady-state temperature reached by the SiPM via a calibration curve. The increase in SiPM temperature due to self-heating is analyzed as a function of the power dissipation in the SiPM and the thermal resistance. This information can be used to adjust the operating voltage of the SiPMs, taking into account the effects of self-heating. Similarly, this approach can be applied to investigate the unknown thermal contact of packaged SiPMs.

## 1. Introduction

Silicon photomultipliers (SiPMs) are the detector of choice for numerous applications in high-energy physics, astroparticle physics, space research, medical imaging, and societal applications. The operating conditions in these applications vary greatly, for example, temperatures can range from cryogenic [1] to 100 °C [2,3]. Usually, devices are operated in the dark, and visible photons are the signal to be detected, but in applications such as LiDAR systems (light detection and ranging), SiPMs need to be operated in the presence of an ambient light background [4,5]. In collider experiments, SiPMs are operated in a harsh radiation environment. The principal effect of radiation damage, already at moderate fluences of $\Phi_{\mathrm{eq}}=10^{10}$ cm^−2^, is an increase in the dark current [6,7]. For the Phase-2 Upgrade, the CMS collaboration at LHC plans two new detectors equipped with SiPMs [8]. In the High Granularity Calorimeter (HGCAL) plastic scintillator tiles with an SiPM readout will equip the rear active layers, which, during the detector lifetime, will accumulate a 1 MeV neutron-equivalent fluence $\Phi_{\mathrm{eq}}=5\times 10^{13}$ cm^−2^ [9]. Even worse will be the operating conditions in the barrel MIP timing detector (MTD), which will reach a fluence of $\Phi_{\mathrm{eq}}=2\times 10^{14}$ cm^−2^ [10]. Understanding the SiPM’s response after radiation exposure is crucial for these applications. Although the dark count rate (DCR) of non-irradiated SiPMs is typically in the range of 10–500 kHz/mm^2^ when operating in a dark environment at room temperature, it has been shown that the DCR increases by five orders of magnitude to above GHz/mm after a fluence $\Phi_{\mathrm{eq}}=10^{13}$ cm^−2^, even with cooling to −30 °C [10,11,12,13,14]. The same high (photo-)current is registered by SiPMs operated with background ambient light. These elevated currents result in significant power dissipation, potentially causing a rise in the SiPM temperature, an effect known as self-heating.

To maintain stable SiPM temperatures in such high dark count rate conditions, it is crucial to ensure a small thermal resistance between the SiPM and the heat sink [15]. Efficient heat dissipation is necessary to prevent self-heating and subsequent variations in the breakdown voltage of the SiPM, which changes its performance parameters such as gain and photon detection efficiency.

This research seeks to devise a method for gauging the temperature increase in an SiPM caused by the dissipation of power in its amplification layer.

The SiPM’s performance depends on the temperature *T*. The photocurrent measured at a constant bias voltage and photon rate decreases with increasing temperature, expressed as $I_{\mathrm{photo}}\propto PDE\left(T\right)\times G\left(T\right)\times ECF\left(T\right)$, where PDE is the photodetection efficiency, *G* is the gain, and ECF is the excess charge factor [16]. The temperature sensitivity of the $PDE\propto FF\times QE\left(\lambda \right)\times P_{\mathrm{Geiger}}\left(V_{\mathrm{bias}},T\right)$ is introduced via the voltage-dependent Geiger-breakdown probability, $P_{\mathrm{Geiger}}$, with FF the device fill factor and QE the wavelength-dependent quantum efficiency. An increase in *T* reduces the carriers’ velocity due to increased scattering on the lattice. The ionization coefficients for holes and electrons decrease with *T*. Consequently, $P_{\mathrm{Geiger}}$ at a fixed $V_{\mathrm{bias}}$ decreases, and the breakdown voltage, $V_{\mathrm{bd}}$, increases. An increase in $V_{\mathrm{bd}}$ with temperature leads to a decrease in gain, described by the relation $G=C_{\mathrm{pix}}\times \left(V_{\mathrm{bias}}-V_{\mathrm{bd}}\left(T\right)\right)/q_{0}$, where $C_{\mathrm{pix}}$ is the pixel capacitance.

In the study by Lucchini et al. [17], a method is proposed to evaluate the heating of SiPMs in various package configurations and to evaluate changes in temperature during operation at elevated dark count rates. The SiPM current ($I_{\mathrm{SiPM}}$) is measured under constant illumination, and changes in $T_{\mathrm{SiPM}}$ are inferred from alterations in $I_{\mathrm{SiPM}}$ when the bias voltage is changed or the thermal conditions are changed by the airflow from a fan. This paper uses a similar approach for deducing temperature variations from current measurements. However, in our method, which we introduced in [18], $I_{\mathrm{SiPM}}$ is measured in a cycle at a constant bias voltage while varying the intensity of light. This prevents abrupt variations in the SiPM working conditions and parameters (*G*, PDE, ENF). The primary assumption underlying our analysis is that $I_{\mathrm{SiPM}}$ under constant illumination is solely dependent on $V_{\mathrm{bias}}-V_{\mathrm{bd}}\left(T\right)$. As part of this study, we cross-checked that both methods, Lucchini’s and ours, measure a consistent temperature increase in the SiPM, independently of whether the power increase is induced by a light intensity or a bias voltage increase.

The investigated SiPMs and the setup are described in Section 2. The experimental method and the data analysis to determine the increase in SiPM temperature are explained in Section 3 and Section 4, respectively. The main results are given in Section 5, followed by the conclusions in Section 6.

## 2. SiPMs Investigated and Experimental Setup

Two types of SiPMs were investigated. First, an R&D SiPM from KETEK, Munich, Germany, [19], type MP15V09, with $V_{\mathrm{bd}}=\left(27.60\pm 0.02\right)V$ at 25 °C [20], pixel size of 15 × 15 ${\mu m}^{2},$ and number of pixels $N_{\mathrm{pix}}=27,367$. The active region is circular with a radius $r_{\mathrm{SiPM}}=1.4\;mm$. This SiPM is manufactured on a 700 μm thick silicon wafer and does not have a specially manufactured entrance window. The bare silicon sensor is glued on an alumina (Al_2_O_3_) substrate, with area 20 × 25 ${mm}^{2}\;a$nd thickness 0.6 mm. The second sample is produced by Hamamatsu Photonics K.K., Hamamatsu, Japan (HPK) [21] MPPC S14160-9769, with an active area of 3.0 × 3.0 ${mm}^{2}\;,$ pixel size 10 × 10 ${\mu m}^{2},$ number of pixels: 89,600 and $V_{\mathrm{bd}}=\left(38.19\pm 0.02\right)V$ at 25 °C. It is delivered as a surface mount package with a silicon resin and a glass–epoxy entrance window and has an overall thickness of 0.8 mm. This sample was tested both soldered on a flexible PCB and on the same alumina as the KETEK sample to compare the different substrates. The information is summarized in Table 1.

The current of the samples, mounted on a temperature-controlled chuck (ΔT≈ 0.1 K), was measured using probe needles connected to a Keithley 6517B voltage source/current meter [22]. The alumina or PCB substrate was fastened to the chuck using a vacuum pump. For good thermal contact, the substrate was placed directly on the surface of the chuck. To emulate a degraded thermal contact of the SiPM, two polyoxymethylene (POM) layers of different thicknesses (1.5 and 3.0 ± 0.1 mm) were placed between the substrate and the cold chuck, increasing the thermal resistance. The thermal conductivity of POM $k=0.3\;Wm^{-1}K^{-1}$ is close to that of commercially available PCB materials. The thermal contact between POM and the Al_2_O_3_/HPK-housing was simply via an air gap and may not have been optimal. Three temperature sensors (PT-100), glued with a thermal foil, were used. Two sensors recorded the temperature on the surface of the substrate, $T_{\mathrm{sensor}1}$ and $T_{\mathrm{sensor}2}$, and a third was on the chuck, $T_{\mathrm{sensor}3}$, as shown in the sketch in Figure 1.

The illumination was provided with light of a wavelength λ=470 nm from an LED operated in DC mode and surrounded by a diffuser. The setup was located in a light-tight and electrically shielded box.

## 3. Method

Two sets of data were recorded. The calibration data set, with the substrate in direct contact with the chuck, $I_{\mathrm{cal}}\left(V_{\mathrm{bias}},T_{\mathrm{chuck}},I_{\mathrm{LED}}\right)$ was taken tuning the LED intensities to different values using the LED current, $I_{\mathrm{LED}}$, with $V_{\mathrm{bias}}$ ranging from below $V_{\mathrm{bd}}$ to about 10 V overvoltage ($OV=V_{\mathrm{bias}}-V_{\mathrm{bd}}\left(T\right)$), and $T_{\mathrm{chuck}}$ varied in five steps of 2.5 °C centered at 25 °C and −30 °C, respectively. From the calibration data, the sensitivity function described in Section 4 was extracted. The sensitivity function was applied to the second set of data, the measurement cycle, used to determine the increase in the SiPM temperature from its measured current under conditions with good and degraded thermal contact.

Figure 2 presents an example of the measurement cycles performed on the KETEK SiPM in three different thermal contact configurations. The SiPM current, $I_{\mathrm{SiPM}}$, is recorded at constant $T_{\mathrm{chuck}}$ and $V_{\mathrm{bias}}$, in three time intervals of 320 s each. This time interval is sufficient for reaching thermal equilibrium. The light intensity is set to zero in the first and third intervals and the dark current, $I_{\mathrm{dark}}$, is measured. In the second interval, the LED light is turned on, which induces a jump in the SiPM current. The effect of self-heating becomes visible in the small decrease in current during step 2. The zoom-in and analysis of this interval are discussed later in Section 4. The light intensity is set by adjusting the current value, $I_{\mathrm{LED}}$, to induce values of dissipated power, $P=I_{\mathrm{SiPM}}\times OV$ in the SiPMs to be in the range of 25 mW to 100 mW. These particular values of dissipated powers were chosen to mimic, on a non-irradiated SiPM, the power observed in the dark on an HPK SiPM after neutron irradiation to a fluence $\Phi_{\mathrm{eq}}=5\times 10^{13}$ cm^−2^ operated at −30 °C and OV = 2.5 V. The *I*–*V* curves used for this consideration are shown in Figure 3. Compared to a non-irradiated SiPM, the irradiated HPK SiPMs demonstrate an increase in the dark current after irradiation by a factor $10^{5}$ at OV = 2.5 V. The same photocurrent is induced on a non-irradiated HPK SiPM by illumination with LED light adjusted to $I_{\mathrm{LED}}$ = 0.05 mA, and operating the SiPM at OV = 2.5 V. It would also be possible for other combinations of $I_{\mathrm{LED}}$ and OV to induce the same power. The photocurrent is defined as the SiPM current minus its dark current: $I_{\mathrm{photo}}=I_{\mathrm{SiPM}}-I_{\mathrm{dark}}$.

## 4. Analysis

From the calibration data, $I_{\mathrm{photo}}^{\mathrm{cal}}\left(V_{\mathrm{bias}},T_{\mathrm{chuck}},I_{\mathrm{LED}}\right)$, the photocurrent sensitivity $S_{\mathrm{photo}}$ is obtained using the following formula:(1)$$S_{\mathrm{photo}}\left(V_{\mathrm{bias}},T_{\mathrm{chuck}}\right)=\frac{1}{I_{\mathrm{photo}}^{\mathrm{cal}}}\times \frac{dI_{\mathrm{photo}}^{\mathrm{cal}}}{dV_{\mathrm{bias}}}\times \frac{dV_{\mathrm{bd}}}{dT_{\mathrm{chuck}}}\left[\frac{1}{K}\right]$$
for different $I_{\mathrm{LED}}$ values. The sensitivity gives the relative change in $I_{\mathrm{photo}}$ for a temperature change of 1 K. In Equation (Equation 1) it is assumed that the sensitivity for a given photon current $I_{\mathrm{photo}}$ only depends on the overvoltage. The sensitivity curves obtained from the calibration data for various values of $I_{\mathrm{LED}}$ as a function of OV are shown in Figure 4 for KETEK (left) and for HPK (right). The HPK measurements were taken at temperatures of 25 °C and −30 °C. In the range of interest of the measurements, the sensitivity is between 0.004 and 0.01
$K^{-1}$. As expected, $I_{\mathrm{photo}}$ increases with $I_{\mathrm{LED}}$ for a constant $T_{\mathrm{chuck}}$, since the current is proportional to the rate of converted photons producing a Geiger discharge ($R_{\gamma }$) and the PDE, $I_{\mathrm{photo}}=q_{0}\times G\times R_{\gamma }\times PDE\times ECF$ [16]. But it can be seen that $S_{\mathrm{photo}}$ almost does not depend on $I_{\mathrm{LED}}$. The effects of heating and occupancy can be ignored when extracting this calibration function. The *T*-dependence of the breakdown voltage is derived from the calibration data recorded at different values of $T_{\mathrm{chuck}}$, yielding to the $dV_{\mathrm{bd}}/dT$ values presented in Table 1.

Once the sensitivity curve is known, the change in SiPM temperature during a measurement cycle (or during operation in an experiment) is calculated from the relative change in photocurrent, divided by the sensitivity:(2)$$\Delta T_{\mathrm{SiPM}}=\frac{1}{S_{\mathrm{photo}}}\times \frac{\Delta I_{\mathrm{photo}}}{I_{\mathrm{photo}}},$$
where $\Delta I_{\mathrm{photo}}=I_{\mathrm{photo}}\left(t_{1}\right)-I_{\mathrm{photo}}\left(t\right)$ is defined as the (time-dependent) difference of the photocurrent to the first measured value after the LED is switched on. If the time $t_{1}$ is sufficiently close to the time of the LED switching, it is assumed that the temperature before and after the switch is still the same. This can be used to impose continuity constraints on the measured temperature profile during a scan. In Figure 5, the temperature changes induced on the SiPM current after switching on the LED in the second interval of the cycles reported in Figure 2 are calculated using Equation (Equation 2). For each cycle, the current $I_{\mathrm{norm}}$ is the SiPM current normalized to the first measurement point of the second interval. The first points of the blue curve show that the main change is in the first step; therefore, the assumption of no T change for the first point after switching on the LED is not completely fulfilled. This may induce the largest error in the calibration of the temperature increase. The temperature of the SiPM mounted directly on the alumina increases only by half a degree, while the SiPMs isolated by POM experiences an increase of 3–4 K, depending on the thickness of the substrate. The oscillations observed in all current measurements are induced by the temperature regulation of the cold chuck, which stabilized the temperature around the set value with a cycle of about 90 s.

The cycle presented in Figure 2 assumes that the SiPM parameters at fixed bias voltage are not instantaneously varied by the increase in light intensity in the second step (B). To evaluate the difference between our assumption and the one applied by Lucchini et al. [17], we performed a second cycle scan, depicted in Figure 6, where the intensity of the LED was kept fixed and the SiPM bias voltage was changed from a value below to one above breakdown, to generate in step B exactly the same power as in the previous cycle.

The temperature changes induced in the SiPM in step B of both cycles of Figure 6 are compared in Figure 7. The values obtained and the temperature gradients are the same for both scans, indicating that the change in SiPM parameters from below to above breakdown does not significantly influence the measurements of self-heating.

In the following section, $\Delta T_{\mathrm{SiPM}}$ values are quoted for times which are long enough that the current has reached a steady state after the switch.

## 5. Determination of Self-Heating

The procedure described in Section 3 and the analysis in Section 4 were applied to investigate the two SiPMs under study, described in Table 1. Four LED light intensities were chosen to study the temperature increase in the SiPMs in the relevant power range expected in the dark after irradiation. The value of 50 mW is marked as a reference in the plots, which corresponds to the power of an HPK SiPM irradiated to a fluence $\Phi_{\mathrm{eq}}=5\times 10^{13}\;{cm}^{-2}$ and operated at OV= 2.5 V and T= −30 °C.

As expected, $\Delta T_{\mathrm{SiPM}}$ is proportional to the power, as demonstrated in Figure 8, for measurements performed at $T_{\mathrm{chuck}}=25\;{}^{\circ}C$.

The best thermal contact is obtained for the SiPM mounted on the alumina substrate and directly placed on the chuck. In this configuration, the increase in temperature calculated using Equation (Equation 2) is $\Delta T_{\mathrm{SiPM}}=1.5\;K$. The 1.5 mm thick POM layer degrades the thermal contact of the SiPM on the alumina substrate to the same level as that of the flexible PCB. For both configurations, $\Delta T_{\mathrm{SiPM}}=3.0\;K$. This demonstrates the significant influence of the thermal contact on the self-heating of SiPMs: the better the thermal contact, the lower the increase in $\Delta T_{\mathrm{SiPM}}$. The points in the plot include the error propagated from the current measurement, but this is too small to explain the deviation of the points from the expected line, which is of the order 5% below 80 mW, and reaches 25% above 80 mW. The data are fit with a proportionality line, which indicates that for dissipated power larger than 80 mW additional systematic effects induce a super-linear temperature increase.

Figure 9 compares the results for the two types of SiPM investigated. At best thermal contact, the minimum temperature increase by self-heating is measured for the KETEK sample, $\Delta T_{\mathrm{SiPM}}=0.2\;K$. This value is considerably lower than that obtained for the HPK sample tested under the same conditions. This may be attributed to the thermal property of the HPK package, which has a higher thermal resistance than the bare thermal contact compared to the bare KETEK silicon sensor. The same is observed when comparing the temperature increase of HPK on PCB with that of KETEK on alumina plus a 1.5 mm thick POM layer, which according to the result shown in Figure 8 should lead to the same thermal increase. The increase in $\Delta T_{\mathrm{SiPM}}$ does not scale linearly with the POM’s thickness, indicating that the thermal coupling via the air gap is not ideal or that a more complex volume effect must be considered. Detailed 3D finite element simulations of the experimental setup are ultimately required to confirm this effect, including the correct boundary conditions.

Table 2 summarizes the $\Delta T_{\mathrm{SiPM}}$ values at 50 mW with the corresponding breakdown voltage increase, $\Delta V_{\mathrm{bd}}$, which is proportional to the increase in temperature and is determined using the $dV_{\mathrm{bd}}/dT$ reported in Table 1.

Measurements taken at 25 °C and −30 °C with the HPK sample soldered on the PCB are compared in Figure 10, and analyzed using Equation (Equation 2). The results are reported in Figure 11.

The $\Delta T_{\mathrm{SiPM}}$ values for 50 mW are summarized in Table 3. The induced temperature increase by self-heating at −30 °C is slightly higher than at 25 °C. Although the relative change in current $\left(\frac{\Delta I_{\mathrm{photo}}}{I_{\mathrm{photo}}}\right)$ is the same at high and low temperatures, the sensitivity at −30 °C ($S_{\mathrm{photo}}=0.6\%/K$) is lower than at 25 °C ($S_{\mathrm{photo}}=0.7\%/K$).

## 6. Conclusions

A method is developed to determine the self-heating of an SiPM from the measurement of its photocurrent. The method is tested as a function of the induced power, thermal contact, and two different substrates on two SiPM models with different packages. Self-heating causes an increase in the breakdown voltage and a decrease in the SiPM photocurrent operated at fixed bias voltage. In these measurements, the SiPM is illuminated by a blue LED, and its current transients are measured at a constant bias voltage when changing the LED current.

The developed method is used to determine the temperature increase for an SiPM with dissipated power around 50 mW, which is the measured power of the HPK SiPM (MPPC S14160-976X, Hamamatsu, Japan) irradiated to a fluence $\Phi_{\mathrm{eq}}=5\times 10^{13}{cm}^{-2}$ and operated at 2.5 V above the breakdown voltage at −30 °C, taken as reference. For different thermal resistances between the HPK SiPM and the chuck, the following results are obtained: for ideal thermal contact (SiPM on Al_2_O_3_ substrate) an increase of between 0.2 K and 1.5 K is observed depending on the SiPM package; for realistic thermal contact (SiPM mounted on a PCB), the temperature is larger and ranges between 2 and 3 K, corresponding to an increase in $V_{\mathrm{bd}}$ of 50 to 100 mV.

These results demonstrate the importance of a proper investigation of the thermal properties of the SiPM package, substrate, and thermal connection to a cooling system when planning to operate at high (dark or photo-) currents. Furthermore, the method proposed in this paper can be used to apply adjustments to the operation voltage depending on the value of the SiPM current during operation.

## Figures and Tables

**Figure 1 sensors-24-02687-f001:**
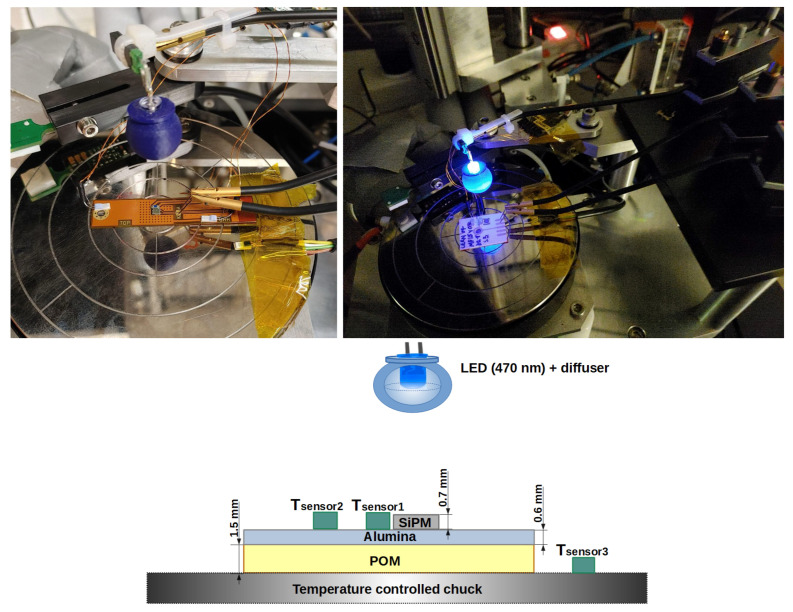
Photos and schematic of the experimental setup. Layout on the probe station: KETEK SiPM glued on the alumina substrate with a POM layer to degrade the thermal contact with the chuck. The distances of the $T_{\mathrm{sensors}}$ on the alumina from the SiPM center are (3.2±0.1) mm ($T_{\mathrm{sensor}1}$) and (7.4±0.1) mm ($T_{\mathrm{sensor}2}$). The alumina substrate is depicted on the right and the PCB substrate is depicted on the left.

**Figure 2 sensors-24-02687-f002:**
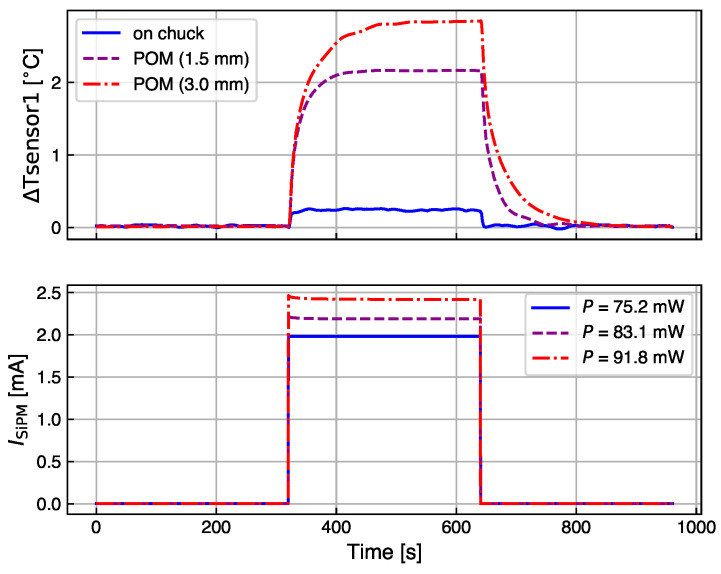
Example of three measurement scan cycles for the KETEK SiPM mounted on alumina (blue) with two POM spacers thicknesses. The temperature change recorded by sensor 1 with respect to the first measurement of the cycle is plotted versus time (**top**) and compared to the SiPM current measurement (**bottom**). The current of the LED is set to zero in the first and third intervals of the cycle. In the second interval, it is switched to a fixed value that is kept constant for the three cycles. The power *P* dissipated by the SiPM in this step of each cycle is reported in the legend. Measurements are taken at 25 °C.

**Figure 3 sensors-24-02687-f003:**
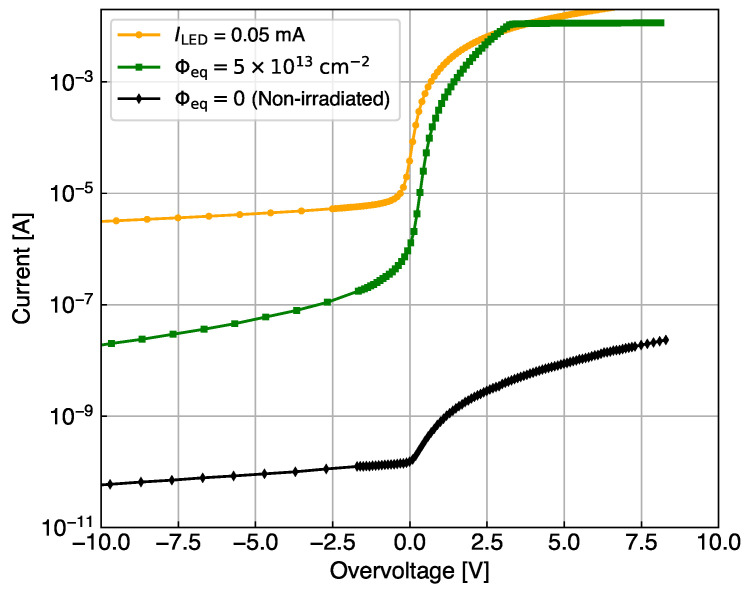
Dark current for a non-irradiated HPK SiPM (black) and the same SiPM irradiated with neutrons at a fluence $\Phi_{\mathrm{eq}}=5\times 10^{13}$ cm^−2^ (green), compared to the photocurrent of the same non-irradiated HPK SiPM exposed to LED light. The light intensity is tuned to reach the same range as the irradiated SiPM at an overvoltage OV = 2.5 V.

**Figure 4 sensors-24-02687-f004:**
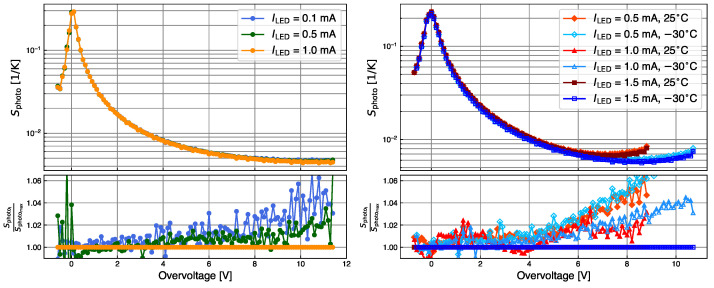
Sensitivity curve at several LED currents for the KETEK sample at 25 °C (**left**), and for the HPK sample (**right**). For HPK data, the red (blue) symbols are taken at 25 °C ( −30 °C). At the bottom, the ratio of sensitivity to that for the largest $I_{\mathrm{LED}}$ is shown.

**Figure 5 sensors-24-02687-f005:**
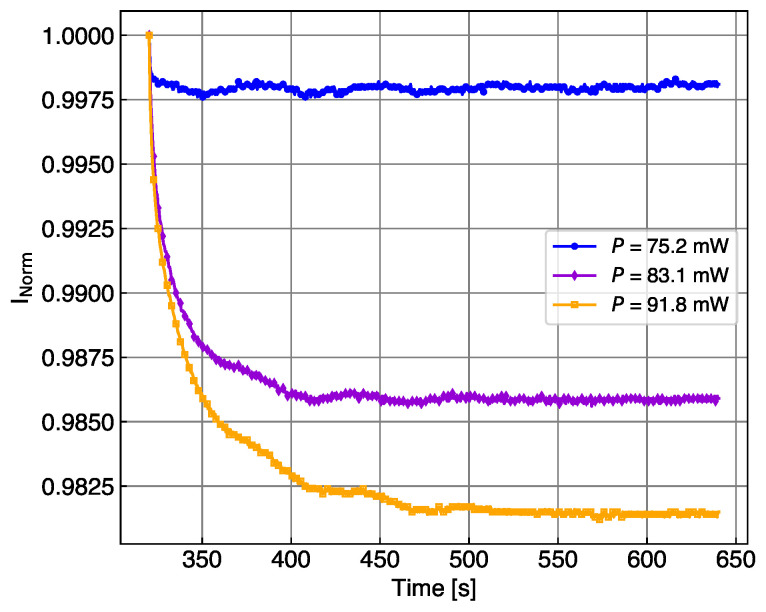
(**Top**) Current of the KETEK SiPM measured at 25 °C in the central interval of the cycles of Figure 2, normalized to the first measurement after the LED switch. The LED light intensity is fixed for the three cycles. The SiPM mounted on alumina thermalizes with a fast time constant, while the SiPM isolated via POM presents a longer thermalization constant. (**Bottom**) Temperature change induced in the SiPM after switching on the LED, calculated using Equation (Equation 2).

**Figure 6 sensors-24-02687-f006:**
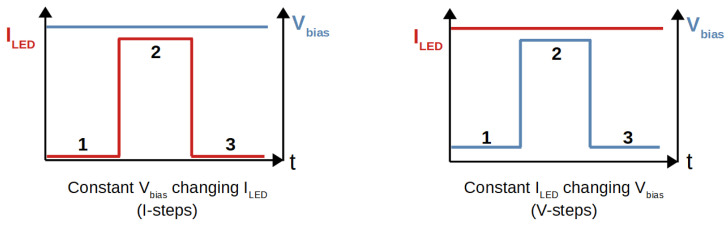
Schematic of the two cycle types performed to obtain the results presented in Figure 7. (**Left**) The LED current is stepped from zero (1) to a tuned value (2) and back (3), while keeping the SiPM bias voltage fixed. (**Right**) The SiPM bias voltage is stepped from 2 V below breakdown (1) to a tuned value (2) and back (3), while keeping the LED current fixed. The tuned values are such that in step B the power in the SiPM is equal in both cycles.

**Figure 7 sensors-24-02687-f007:**
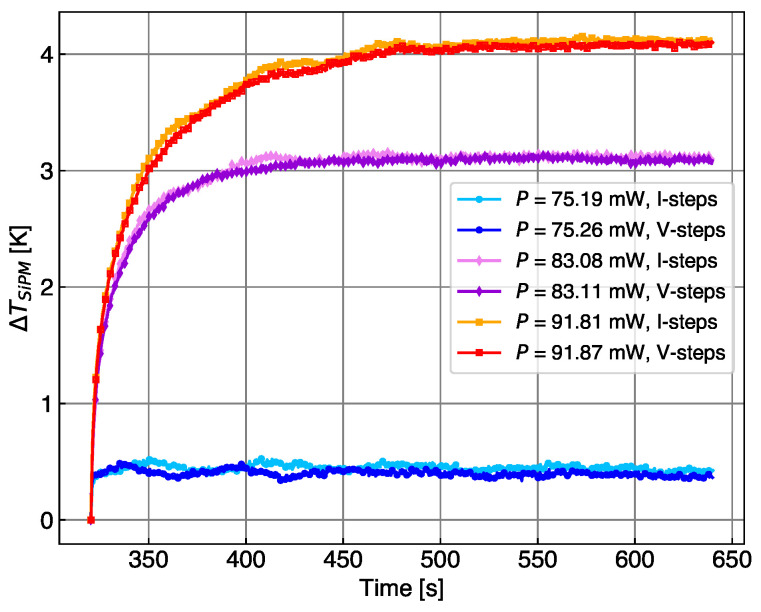
Temperature change induced in the SiPM in step B of the two cycles discussed in Figure 6.

**Figure 8 sensors-24-02687-f008:**
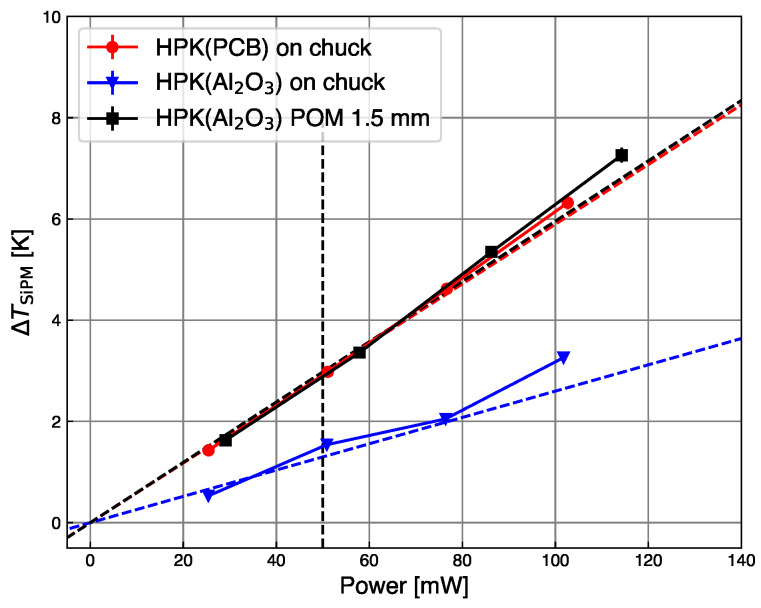
Temperature increase by self-heating as a function of the power dissipated in the SiPM. Best thermal contact is obtained with the alumina substrate directly on the chuck (blue triangles). The thermal contact of the flexible PCB (red circles) shows the same performance as the alumina substrate with a 1.5 mm thick POM spacer (black squares). The maximum deviation of the points from the linear fit (dash lines) is 25% above 80 mW. The vertical dash line at 50 mW indicates the values reported in Table 2.

**Figure 9 sensors-24-02687-f009:**
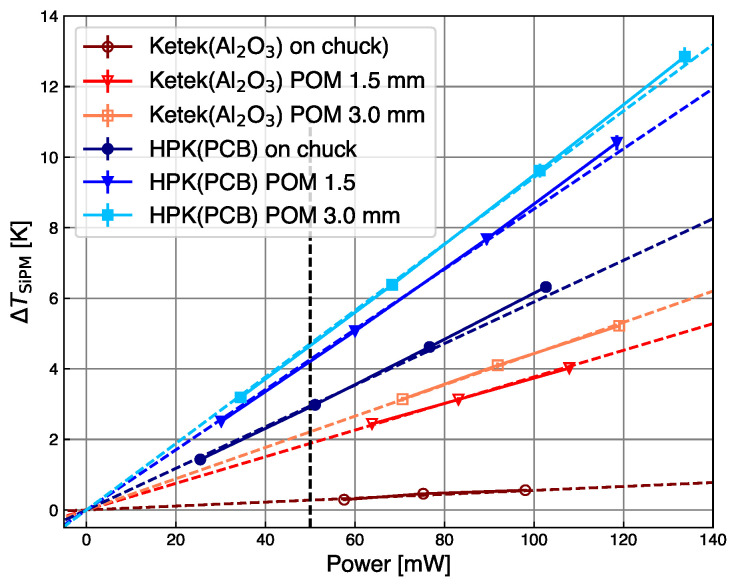
Comparison of temperature increase by self-heating as a function of the power dissipated for two SiPM models: KETEK (red), and HPK on PCB (blue). Circles represent data taken with the substrate on top of the chuck surface, triangles are for data using the thinner POM layer in between, and squares for the thicker POM layer. The dotted lines are linear fits to the data. The vertical dashed line corresponds to the reference power of 50  mW. The maximum deviation of the points from the linear fit (dashed lines) is 4% above 100 mW. The vertical dash line at 50 mW indicates the values reported in Table 3.

**Figure 10 sensors-24-02687-f010:**
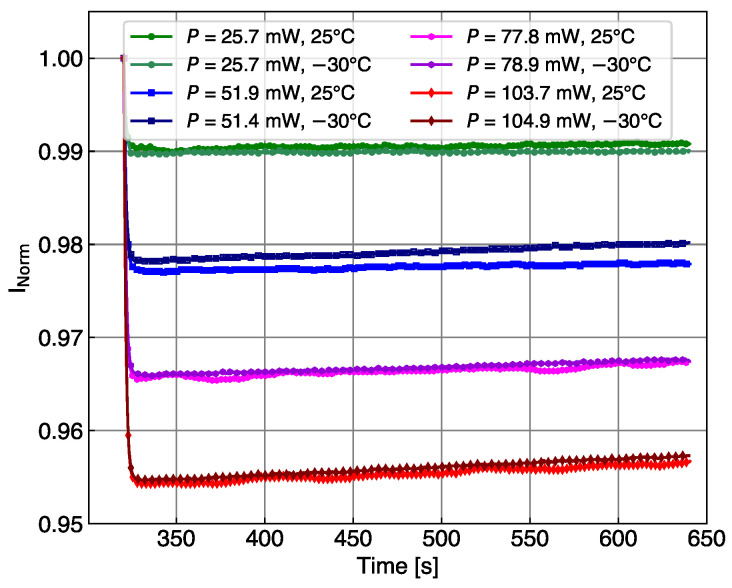
SiPM current measured in step B of the cycle performed with the HPK SiPM at 25 °C and at −30 °C. The oscillations in the current induced by the automatic temperature stabilization of the chuck are less pronounced at low temperatures. This is a feature of the ATT-system controller.

**Figure 11 sensors-24-02687-f011:**
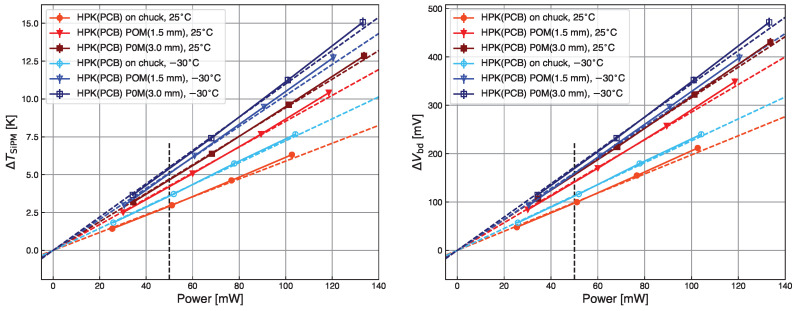
Comparison of temperature increase by self-heating (**left**) and breakdown voltage shift (**right**) as a function of the power dissipated for 25 °C (red) and −30 °C (blue). Circles indicate data taken with the substrate on top of the chuck surface, and triangles (squares) data using the thinner (thicker) POM layer. The dotted lines are linear fits to the data. The vertical dashed line corresponds to 50 mW.

**Table 1 sensors-24-02687-t001:** Summary information of the two SiPM models used in this study.

Sample ID	S6	S14160-9769
Manufacturer	KETEK	HPK (MPPC)
# pixels	27,367	89,600
Pixel size [μm]	15	10
Active area [$mm^{2}$]	6.2	3.0×3.0
Sample thickness [μm]	700	800
Substrate	Alumina	PCB/Alumina
$V_{\mathrm{bd}}$ (25 °C) [V]	27.6	38.2
$V_{\mathrm{bd}}$ ( −30 °C) [V]	-	36.3
$dV_{\mathrm{bd}}/dT$ [$mV\;K^{-1}$]	22.4	33.5 (RT)
$dV_{\mathrm{bd}}/dT$ [$mV\;K^{-1}$]	-	31.3(low T)
$S_{\mathrm{photo}}$ [%/K] (25 °C)	0.46%(10.4 OV)	0.69%(6.8OV)
$S_{\mathrm{photo}}$ [%/K] ( −30 °C)	-	0.60%(7.7OV)

**Table 2 sensors-24-02687-t002:** Breakdown voltage and temperature increase by self-heating for KETEK and HPK SiPMs at 25 °C dissipating 50 mW power. The overvoltages during the measurement cycles were OV=10.4 V (KETEK), OV=6.8 V (HPK). The errors are estimated to be around 5%.

Sample	Substrate	POM [mm]	$\Delta V_{\mathrm{bd}}$ [mV]	$\Delta T_{\mathrm{SiPM}}$ [K]
KETEK	Al_2_O_3_	0	6	0.2
KETEK	Al_2_O_3_	1.5	44	2.0
HPK	Al_2_O_3_	0	50	1.5
HPK	Al_2_O_3_	1.5	100	3.0
HPK	PCB	0	100	3.0
HPK	PCB	1.5	140	4.2
HPK	PCB	3.0	154	4.6

**Table 3 sensors-24-02687-t003:** Breakdown voltage and temperature increase by self-heating HPK SiPM at 25 °C and −30 °C dissipating 50 mW power. The overvoltages during the measurement cycles were OV=6.8 V (HPK, 25 °C), and OV=7.7 V (HPK, −30 °C). The errors are estimated to be around 5%.

Sample	Substrate	POM [mm]	*T* [°C]	$\Delta V_{\mathrm{bd}}$ [mV]	$\Delta T_{\mathrm{SiPM}}$ [K]
HPK	PCB	0	25	10	3.0
HPK	PCB	0	−30	113	3.6
HPK	PCB	1.5	25	140	4.2
HPK	PCB	1.5	−30	160	5.1
HPK	PCB	3.0	25	158	4.7
HPK	PCB	3.0	−30	170	5.4

## Data Availability

Data will be made available on request.

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
