# Peer review of "Determination of Self-Heating in Silicon Photomultipliers"

_sensors, 2024, doi:10.3390/s24092687_

Round 1

Reviewer 1 Report

Comments and Suggestions for Authors

The manuscript compares two methods of evaluating, monitoring and compensating self-heating effects in Silicon Photomultipliers. One of the methods compared was previously introduced by authors of this manuscript. The manuscript presents a novel research, appropriately designed and described and confirms the consistency of the two methods for evaluating of SiPM self-heating. 

Before acceptance these issues must be resolved: 

-Title: the current title is not fully describing the contents. The manuscript deals with methods of monitoring temperature due to self-heating, rather than investigating the reasons of mechanisms of self-heating. Modify the title to take this into account

- Abstract: State more clearly (towards the end of the abstract) what is the concrete result of this study

- Introduction, L18 - please add more relevant citations to cover the statement "temperatures can range from cryogenic to 100 C"

- Figure 1 - almost identical Figure is presented in the previously published NIMA paper, arxiv 2205.05504. Please check if you need to obtain permission of the Elsevier to use this figure

L111-115 The method description is unclear. Maybe try to break into shorter sentences.

L112 was value of 75 mW used in the results of Figure 2? If so, state that clearly.

Labeling: I_SiPM, I_LED, I_dark, I_photo are appearing throughout the text. Although the names may be self-explanatory, you must define these quantities explicitly. Also make sure you are using the labels consistently throughout the text.

L118 "OV" is unusual. I suggest V_OV

L118 "fresh" -> "unirradiated" and also throughout the text.

Figure 3. It is not clear why data for KETEK and HPK are compared under different conditions. You should present the data for the two SiPMs under same conditions otherwise I do not see the point in this plot. Also the label V_ex is not consistent with previous for overvoltage. Title of the plot should be removed and contained in the caption.

L127 I_photo is not "photon flux", although they are highly correlated

Figure 4. I_LED. "_LED" must be in subscript ILED

L134 does not depend -> I would say "almost does not depend"

Figure 5. Define I_norm

L177 and L179. You must add uncertainties to the quoted values of DeltaT

Author Response

Dear Reviewer,

We would like to express our gratitude for the careful reading and the many comments that helped us to improve the quality of our paper, hopefully to your satisfaction. We include the file with the reply to all reviewer's comments and the list of all changes implemented to the paper.

Reviewer 2 Report

Comments and Suggestions for Authors

The paper presents a study on the self-heating effects in Silicon Photomultipliers (SiPMs) caused by the increase in the dark current due to radiation damage. Measurements were performed using two different SiPMs mounted on either Al2O3 or a flexible PCB substrate. The SiPMs were mounted on temperature-controlled chucks or layers with known thermal resistance. SiPMs were illuminated using LED to determine the steady state temperature. Based on this information, the effects of self-hating can be taken into account for adjusting the operating point of the device. I recommend the acceptance of the paper, but only after a major revision.

Major objections and comments:

1.       While the paper investigates interesting self-heating effects in the SiPMs in relation to possible radiation-induced damage and adjusting the operating point of the device, the results and analysis overlap to some degree with the results presented by the same authors in a previous paper (Self-heating Effect in Silicon-Photomultipliers).

2.       The thermal contact between the chuck and the sample depends on the surface of the chuck and the sample, and the applied pressure (vacuum). Can you estimate the value of the thermal conductivity between the chuck and the sample, and does it affect the analysis of the self-heating of SiPMs? What about the thermal contact between POM and Al2O3 housing established via an air gap?

3.       What is causing the increase in I_SiPM shown in Fig. 2 (bottom) at the beginning of interval 2?

4.       What is the meaning of the values of P written in the legend of Fig 2 (bottom)? How was P calculated and defined?

5.       The comparison of two different devices, an illuminated KETEK, and a non-irradiated/irradiated HPK shown in Fig. 3, is difficult to follow. Why is the comparison given in the first place for different devices? Why is the LED current chosen to achieve the same device current of illuminated and irradiated devices at different overbias voltages? On another point, the legend in Fig. 3 is difficult to follow, and the reader has to carefully check the caption of the text to understand what is shown in the figure.

6.       What is causing the oscillations in the top and bottom curves in Fig. 5?

7.       In the plots of deltaT_SiPM as a function of dissipated power (Fig. 8 and Fig. 9), the linearly extrapolated curves cross the 0 mW or 0 K axes rather randomly. What is happening with self-heating at low powers of dissipation? Does Equation 2 work for the low dissipated power for SiPMs?

8.       For a chuck temperature of -30 °C there are no transient oscillations (or they are less visible) of I_norm as shown in Figure 10. Please explain what is causing this behavior.

Other issues:

1.       The title of the paper is too general.

2.       How is deltaTsensor1 defined in the top panel of Figure 2?

3.       Please define I_norm introduced in Fig. 5. in the text.

4.       The authors interchange the definition and variable for excess bias voltage. In Figure 3, the X-axis shows V_ex, while in the caption it is written OV. Also, sometimes the excess bias voltage is referred to as overvoltage (page 3 line 102, page 3 line 115, Fig 4 caption, Table 2 caption, Table 3 caption) and sometimes as excess bias voltage (Fig 3 caption, page 5 line 128, page 4 line 118). This introduces unnecessary ambiguity in the text and impacts the readability. Be consistent with the definitions and variables used in the text.

5.       Only in Figure 6, do the authors introduce the cycle naming scheme (A, B, C) and explain the SiPM biasing and LED current modulation. In the text describing Fig. 2, the cycles are enumerated as ‘intervals’. This again introduces ambiguity in the manuscript and affects the clarity of the paper. Please rewrite to make it more understandable to the readers (e.g., put the current Fig. 6 before the current Fig. 2 or make other modifications to the order of the text).

6.       Page 2, line 54: Define all the variables that are used in the manuscript. What is C_pix?

7.       The legend in Fig. 10 is difficult to follow as it overlaps with the thick lines in the figure. 

Comments on the Quality of English Language

There are some grammatical errors in the text. Please thoroughly proofread the paper and fix the minor language issues.

Author Response

(The authors gave the same response as above.)

Reviewer 3 Report

Comments and Suggestions for Authors

The paper illustrates a technique to measure the self-heating od SiPMs, a topic that is interesting for the R&D connected to several future experiments in high energy physics and other fields. The method is clearly described and the results on a few samples are provided.

The solution proposed is simple, effective and can be really useful when delicate decisions are required. Although not extremely innovative, the contribution is valid and should be published as it is. 

Author Response

We thank the reviewer for the positive comments.

Reviewer 4 Report

Comments and Suggestions for Authors

This research seeks to devise a method for gauging the temperature increase in SiPM caused by the dissipation of power in its amplification layer. The idea is interesting but need further consideration before publication.

Q1:There are some typos in the manuscript and the text needs to be edited grammatically.

Q2:In Figure 2, the legend for SiPM current measurement (bottom) is unclear. Is it depicting the driving power of the LED light or the dissipated power generated by the SiPM? How is it related to the text description on line 112? Could the authors clarify this?

Q3:At line 122, " The photo-current is defined as Iphoto(ILED) = ISiPM(ILED) – lSiPM(ldark)." , is the author equating Iphoto with ILED? If so, I think this will cause ambiguity to the reader. Could the authors please revise this or clarify it in the manuscript?

Q4:There are quite a few descriptive errors in the manuscript that should not be there, for example, in Figure 8 "Best thermal contact is obtained with alumina substrate directly on the chuck (green squares). The thermal contact of the flexible PCB (red circles) shows the same performance as the alumina substrate with 1.5mm thick POM spacer (green triangles). " I don't see "green triangles". I don't see "green squares" and "green triangles". I suggest the authors to revise the manuscript carefully and resubmit it.

Comments on the Quality of English Language

 Minor editing of English language required

Author Response

(The authors gave the same response as above.)

Round 2

Reviewer 2 Report

Comments and Suggestions for Authors

No further comments.

Reviewer 4 Report

Comments and Suggestions for Authors

The paper has been revised according to the revision suggestions.